# Hi-C implementation of genome structure for *in silico* models of radiation-induced DNA damage

**Samuel P. Ingram** [1,2]☯ *, **Nicholas T. Henthorn** [1,3]☯, **John W. Warmenhoven** [1,3], **Norman F. Kirkby** [1,3], **Ranald I. Mackay** [1,2], **Karen J. Kirkby** [1,3], **Michael J. Merchant** [1,3]

**1** Division of Cancer Sciences, Faculty of Biology, Medicine and Health, The University of Manchester, United Kingdom, **2** Christie Medical Physics and Engineering, The Christie NHS Foundation Trust, Manchester, United Kingdom, **3** The Christie NHS Foundation Trust, Manchester Academic Health Science Centre, Manchester, United Kingdom

☯ These authors contributed equally to this work.
* samuel.ingram@postgrad.manchester.ac.uk

**Data Availability Statement:** All datasets are available at https://data.mendeley.com/datasets/kzycj3n2mm. The G-NOME source code is

## Abstract

Developments in the genome organisation field has resulted in the recent methodology to infer spatial conformations of the genome directly from experimentally measured genome contacts (Hi-C data). This provides a detailed description of both intra- and inter-chromosomal arrangements. Chromosomal intermingling is an important driver for radiation-induced DNA mis-repair. Which is a key biological endpoint of relevance to the fields of cancer therapy (radiotherapy), public health (biodosimetry) and space travel. For the first time, we leverage these methods of inferring genome organisation and couple them to nano-dosimetric radiation track structure modelling to predict quantities and distribution of DNA damage within cell-type specific geometries. These nano-dosimetric simulations are highly dependent on geometry and are benefited from the inclusion of experimentally driven chromosome conformations. We show how the changes in Hi-C contract maps impact the inferred geometries resulting in significant differences in chromosomal intermingling. We demonstrate how these differences propagate through to significant changes in the distribution of DNA damage throughout the cell nucleus, suggesting implications for DNA repair fidelity and subsequent cell fate. We suggest that differences in the geometric clustering for the chromosomes between the cell-types are a plausible factor leading to changes in cellular radiosensitivity. Furthermore, we investigate changes in cell shape, such as flattening, and show that this greatly impacts the distribution of DNA damage. This should be considered when comparing *in vitro* results to *in vivo* systems. The effect may be especially important when attempting to translate radiosensitivity measurements at the experimental *in vitro* level to the patient or human level.

available at https://gitlab.com/PRECISE-RT/
releases/g-nome.

**Funding:** K.J.K, R.I.M, N.F.K and M.J.M were
funded by the NIHR Manchester Biomedical
Research Centre [grant number: BRC-1215-
20007]. S.P.I, M.J.M, K.J.K and R.I.M were funded
by the STFC Global Challenge Network+ in
Advanced Radiotherapy [grant number: ST/
N002423/1]. N.F.K, K.J.K, R.I.M, M.J.M, N.T.H and
J.W.W were funded by the European Union
Horizon 2020 Research and Innovation [grant
number: 730983 – INSPIRE]. K.J.K, R.I.M, N.F.K
and M.J.M were funded by the United Kingdom
Engineering and Physical Science Research
Council [grant No.: EP/S024344/1 – BioProton]. S.
P.I was funded by The Christie Charitable funds -
The Christie NHS Foundation Trust (https://www.
christie.nhs.uk/the-christie-charity). The funders
had no role in study design, data collection and
analysis, decision to publish, or preparation of the
manuscript.

**Competing interests:** The authors have declared
that no competing interests exist.

## Author summary

We have used a technique which allows us to understand how parts of our DNA are orga-
nised within a cell nucleus. This technique has previously shown differences in the organi-
sation between different cell-types. In this study, we show that these differences produce
significant change in the way our DNA is damaged when exposed to radiation. This is
important to understand as one of the primary ways we treat cancer is using radiotherapy.
However, whilst we attempt to target the cancer with radiation, some healthy tissue also
receives radiation. It is the radiation delivered to the healthy tissue which limits how
much radiation we can safely give to the cancer without causing significant side effects in
patients. To know how much radiation we can give, over time, we have learnt generally
safe amounts of radiation that can be given to healthy tissue. Even so, sometimes patients
will still have worse side effects than what we would have predicted. If we want to further
improve our treatments and patient safety, we need to better understand how this safe
limit varies between each patient. The first step in to fully understanding this process
comes from a better understanding of how different cell-types are affected by radiation,
which is partly driven by DNA organisation, shown in this work.

## Introduction

The research fields of radiobiology and DNA structure have shared a symbiotic past. Radiation
has been used to infer the presence of chromosome territories [1], structural cytotoxic
responses [2, 3] and examinations of the chromatin dynamics [4]. In turn, as we gain a better
description of the DNA and chromatin structure we observe an intrinsic relationship with the
radiobiological properties of a cell [5, 6]. This is due to the radiobiological response being
majorly driven by damage to the DNA structure. The formation of double-strand breaks
(DSBs), which is where the sugar-phosphate backbone of the DNA is broken on both sides in
close proximity (<10 bp) has been shown to correlate strongly with cellular survival [7, 8].
This is thought to be partially caused by DSBs giving rise to the possibility of chromosomal
interchanges, whereby chromosomes that misrepair can form a whole variety of chromosome
aberrations [9]. It is believed that a major factor for misrepair events is the mobility and spatial
distribution of DNA break ends [10–12], increasing the importance of chromosomal arrange-
ment on cell fate. One of the more lethal chromosome aberration types are those that involve
two different chromosomes being misrepaired and plays a major role in the radiation-induced
cell death [13, 14]. The probability for these types of chromosomal translocation to occur has
been shown to relate to the intermingling of the different chromosomes [15]. Therefore, there
is an innate interest in being able to accurately predict the spatial distribution of interchromo-
somal DNA break ends as this will inform us to the probability of inducing interchromosomal
aberrations.

 The field of chromosome organisation has made major strides, many of which surround
the microscopy technique fluorescence *in situ* hybridization (FISH). However, this cytogenetic
approach is ultimately limited by its sensitivity and resolution. The field has culminated in the
emergence and development of inference based methods to examine genome organisation
called chromosome conformation capture techniques [16]. Chromosome conformation cap-
ture uses DNA cross-linking to capture interactions between proximal regions of the genome,
the frequencies at which sections of the genome are captured proximal to one another gives a
mechanism by which to infer spatial proximity. One such method is Hi-C, which uses high-
throughput sequencing to capture proximal regions for the entire genome. The Hi-C

technique has been rapidly developing, from initial work of experimental design [17] through to the complex inferences, on both nuclear structure and function [18–22], which can be made from the gathered data [23, 24]. Whilst, the field is still developing it has become clear that an area of interest is the inference of the three-dimensional genome structure from the contact probabilities observed in Hi-C experiments. This has been attempted using multi-dimensional scaling [24], polymer [23, 25–27] and statistical [28, 29] type models with a varying structural focus.

On the other side, radiobiological models of DNA damage and repair are a concept that have been developing for several decades to better understand the effects of radiation in areas such as, healthcare, public health and space travel. Some of these models have proved themselves critical for informing the use of clinical radiotherapy [30, 31]. The clinical efficacy of radiotherapy treatment is limited by the compromise between cell kill in the tumour and in normal tissue, resulting in tumour control and normal tissue complication respectively [32]. Radiobiological models aim to provide insights here, with the intention of better informing the clinician when confronted with this compromise through exploitation of the 5 R's of radiobiology (repair, repopulation, reoxygenation, redistribution and radiosensitivity) [33]. In a subset of radiobiological models, radiation track structure is simulated with an interpretation of the genome structure [34–36], providing a detailed representation of the energy deposition at the DNA level. These structural interpretations have varying levels of complexity but are most commonly relatively uniform and mathematically driven. Furthermore, if cell-type specific geometric models are used, they will generally be the same for all simulations [37] absent of any cellular variation that is found in populations of cells. This variation is prevalent within the radiobiological experimental results and creates ambiguity when interpreting data for clinical decision making [38, 39]. The results of DNA damage models can be used in radio-response models [40] which try to identify effects at the cellular level [12, 41–43]. Ultimately, radiobiological models aim to describe cellular radio-response, with the hope of transference to patient-level response to better predict outcomes of radiation-based treatments.

In this study, we extend the overlap of these research areas by incorporating the chromosome structure, in the form of Hi-C data, in Monte Carlo radiation track structure simulations of DNA damage. This improved geometric representation is crucial to accurately model types of chromosome aberration and the subsequent effects of cell death, senescence or possible radiation-induced mutations. These biological endpoints are key drivers of clinical outcome following radiotherapy that will be benefited by an improved understanding of their origin. To evaluate the effects of a Hi-C geometry we have developed G-NOME (G-NOME—Nuclear Organisation Modelling Environment) to infer the geometry from the Hi-C data. G-NOME is a highly extensible python library that allows for geometry inference using a Markov-chain Monte Carlo polymer model with the ability to be directly fed into tool-kits of radiation track-structure, such as Geant4-DNA [44] and TOPAS-nBio [45]. Through the combination of inferred genome structure from experimental Hi-C data and radiation track structure, it is possible to better encapsulate and therefore understand some of the biological variation seen within experimental radiobiology. Furthermore, this study has highlighted that Hi-C derived geometries for different cell lines have varying amounts of chromosomal intermingling, which are likely to be fundamental drivers for differences in observed cellular radiosensitivity.

## Results

### Definition of terms

In the purpose of these definitions an object can refer to either polymer beads (when describing geometry distribution) or DSBs (when describing DNA damage distribution).

Cluster radius: the spherical radius of inclusion space which counts the objects (e.g. beads or DSBs) within it. clustering: the number of objects that fall within a given radius averaged for all objects in the simulation. Interchromosomal clustering: the number of objects that fall within the cluster radius and *are not* on the same chromosome (including homologous chromosomes) averaged for all objects in the simulation. Intrachromosomal clustering: the number of objects that fall within the cluster radius and *are* on the same chromosome (not including homologous chromosomes) averaged for all objects in the simulation. Inter/Intra-chromosomal clustering ratio: the ratio of the averaged interchromosomal clustering and the intrachromosomal clustering for the object being examined.

## Effects of using LADs and ellipsoid nuclei in Hi-C geometries

The IMR90 (human fetal lung fibroblast) variant conformations, inclusion of Lamina-associated domains (LADs) and flattening of the nucleus to form an ellipsoid, were solved using the G-NOME software and viewed for gross abnormalities (Fig 1A). For each of the three tested cell types, 200 geometries were created. All ellipsoid and Lamina-associated domain (LAD) geometries were solved for 4 million iterations using the G-NOME software, this was chosen to achieve comparable nucleus-bead outliers compared to the spherical geometry, which were solved for 2 million iterations (Fig 1B).

To examine differences in interchromosomal proximity within the modelled cell nucleus the bead clustering of each cell variant is plotted (Fig 1C) for varying cluster inclusion radii. The single value of interchromosomal bead clustering is the result of analysing every bead in the model for the number of interchromosomal beads that are within the tested cluster radius of the bead being analysed, the per bead clustering value is then averaged to obtain a per cell geometry bead clustering. This allows for a coarse examination of differences in the "intermingling" between chromosomes for each of the cells. When incorporating LAD objectives in the geometries there is a slight increase in the amount of intermingling of the chromosomes. Whereas, when the geometry is solved as an ellipsoid the intermingling decreases.

The optimised distribution of the chromosomes are reviewed by according to the positioning of their constituent beads, scored between central or peripheral of the modelled nucleus (Fig 1D). Larger chromosomes (chr1-chr9) are predominately situated at the periphery and smaller chromosomes (chr13-chrX) are situated centrally. This is similar to the chromosomal ordering seen in other models [23]. The analysis shows an increased peripheral and decreased peripheral bead positioning for IMR90 LADs and ellipsoid respectively when compared to the standard IMR90 variant (no LADs and spherical). Although all analysis methods halved the volumes to define the central and peripheral regions, in the ellipsoid geometry it becomes harder for the optimiser to place beads within the periphery that doesn't incur a cost due to the constraint placed on fitting beads within the nucleus. This promotes central placement of the beads within the cell nucleus as indicated by the percentage of DNA content placed in the periphery in Fig 1D.

To identify which chromosomes have beads which are consistently proximal across multiple inferences of chromatin arrangements, a series of chord plots (Fig 1E) were generated from the 200 cell geometries. Each linking line represents chromosomes that share at least one proximal (<500nm) interchromosomal bead in at least X% of the examined 200 geometries, where X can be a particular threshold. When analysing consistent chromosomes with proximal interchromosomal beads, there are no such examples which occur within 50% of the ellipsoid sample analysed. Therefore, all variants have been analysed at a 40% threshold to examine the difference between the conformations. There are more chromosomes that share proximal interchromosomal beads in the spherical geometries than the ellipsoid geometry. Interestingly,

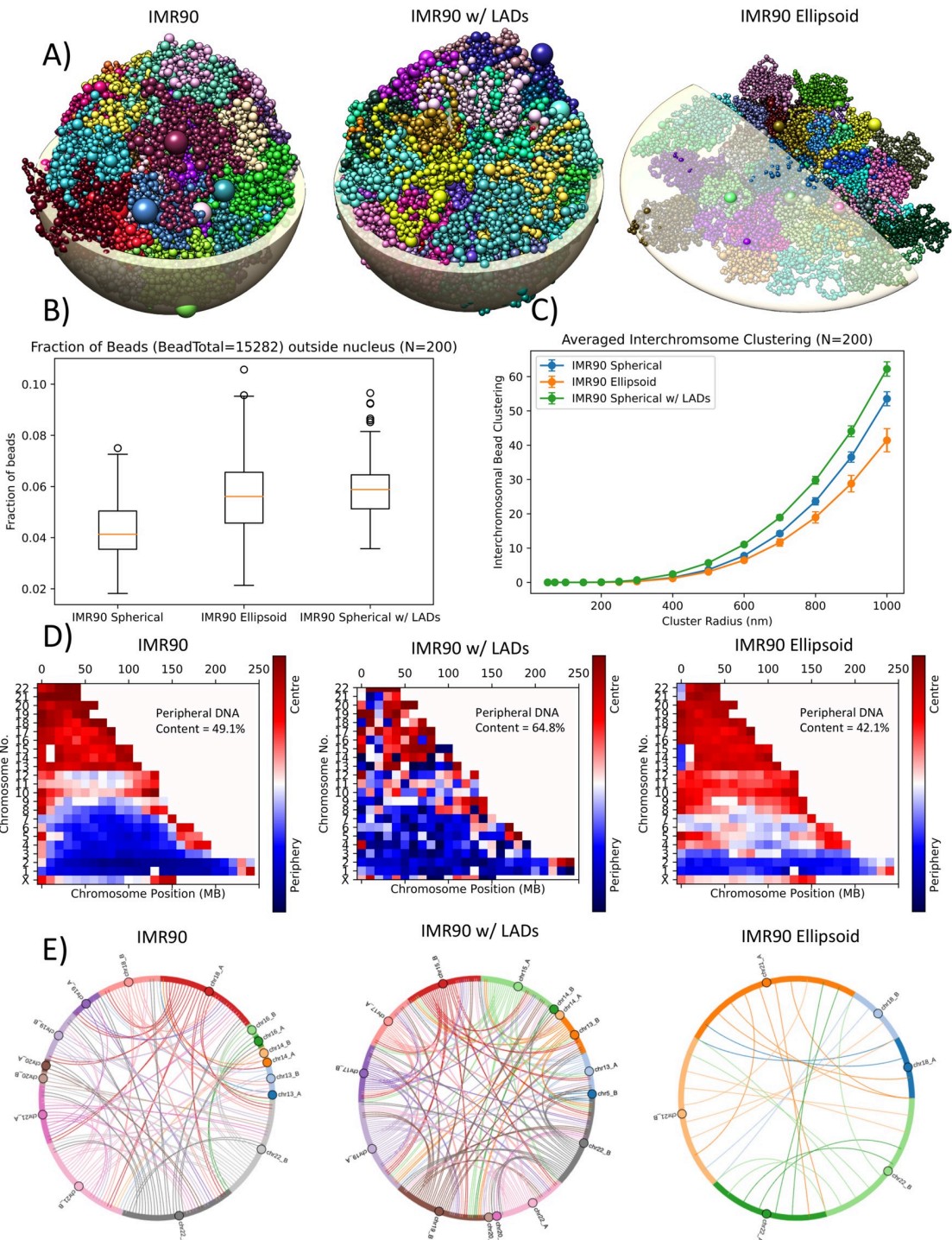

**Fig 1. Geometrical properties of an inferred encapsulation of the 3D spatial chromatin arrangement. A)** rendered 3D example of IMR90 Hi-C data solved for variants of adding LADs or optimising for an ellipsoid geometry. Each colour represents different chromosomes and the bead size represents the amount of DNA content. Rendered nuclear shell is to give an idea of scale for a 5 $\mu$m radius spherical nucleus for IMR90 and IMR90 w/ LADs, the ellipsoid nucleus is shown for 1.0x11.8x11.8$\mu$m. **B)** comparison of the fraction of beads that were not encapsulated by the optimiser objective nucleus. **C)** interchromosomal bead cluster analysis for the IMR90 variants. **D)** chromatin positioning analysis for beads that have been scored based on being within the central or peripheral half of the nuclear volume and is averaged over a 10 Mbp bin. The overall percentage of DNA content placed in the periphery is displayed within each plot. **E)** chromosomes that have interchromosomal beads within a 500 nm radius of one another. Lines are only displayed for the chromosomes that share proximal beads in at least 40% of the examined geometries. The data shown is averaged for 200 inferred geometries for each of the cell variants. Error bars are the standard deviation of 200 geometries.

the chromosomes included in the chord plot for the ellipsoid geometries occur in the spherical geometries also, suggesting these are interaction driven from the experimental Hi-C contact maps rather than compromises due to other bead constraints. There are the same number of chromosomes that share proximal interchromosomal beads for both the geometries with and without LADs. However, there is an increase in the chord density with the inclusion of LADs, suggesting an increased amount of consistent intermingling at the analysed 500 nm radius, which is reasonable given the increase in constraints for these geometries.

## Variations in Hi-C geometries for different cell-type

All spherical cell-type geometries (IMR90—human fetal lung fibroblast, HMEC—human adult mammary epithelial and GM12878—human B-lymphocyte), were solved using the G-NOME software with an optimisation limit of 2 million iterations of successful movements. For each of the three tested cell types, 200 geometries were solved. The resultant conformations were visually examined to check for gross abnormalities in the same manner as the cell variants (Fig 2A). To examine the optimisation of the three cell-type Hi-C datasets, which have different number of constraints, the average cost per constraint for the geometries are plotted for comparison (Fig 2B). Whilst there is quite a large variation observed within the same cell type, there is an overlap between the interquartile ranges of the different cell types, suggesting suitable optimisation of the different datasets and allowing direct comparison.

Cell type changes in the interchromosomal bead clustering are shown in Fig 2C. It can be observed that the cell line GM12878 (human B-lymphocyte) had increased levels of intermingling than the other two cell lines.

The spatial positioning analysis (Fig 2D) shows the same resultant distributions largely follow the same pattern between the different cell-variants. Larger chromosomes (chr1-chr9) are predominately situated at the periphery and smaller chromosomes (chr13-chrX) are situated centrally.

The cell types were analysed for consistent chromosomes with proximal interchromosomal beads (Fig 2E) in the same manner as the cell variants. However, due to the increase of consistent shared proximal beads in comparison to the ellipsoid geometry the threshold was increased to 50% to better compare amongst the three cell types. The inferred GM12878 geometries share a higher amount of proximal chromosomes than both IMR90 and HMEC. The observed number of chromosomes that share a 500 nm proximity matches the observed amount of interchromosomal bead clustering, highlighting that this difference in Fig 2C is due to having an increased number of intermingling chromosomes. The chromosomes that, on average, have mutual proximal beads are those of the smaller and centrally located chromosomes observed in Fig 2D.

## Comparison of Hi-C genome organisation and pseudo-random organisation

To better quantify the genomic organisation of solved Hi-C geometries we compared the interchromosomal bead clustering of the different cell-types and variants against a set of pseudo-random geometries (Fig 3A). These pseudo geometries have no bead-bead constraints and can be placed anywhere within the specified nuclear volume (5 $\mu$m radius). Through the removal of organisation constraints the geometries become devoid of chromosome territories, resulting in the upper achievable level of chromosome intermingling (Fig 3B). The corresponding bead clustering values from the pseudo-random geometries were used to obtain a normalised value for all the Hi-C cell-types and variants (Fig 3C). The lack of chromosome territories results in a linear log-log relationship for the pseudo-random

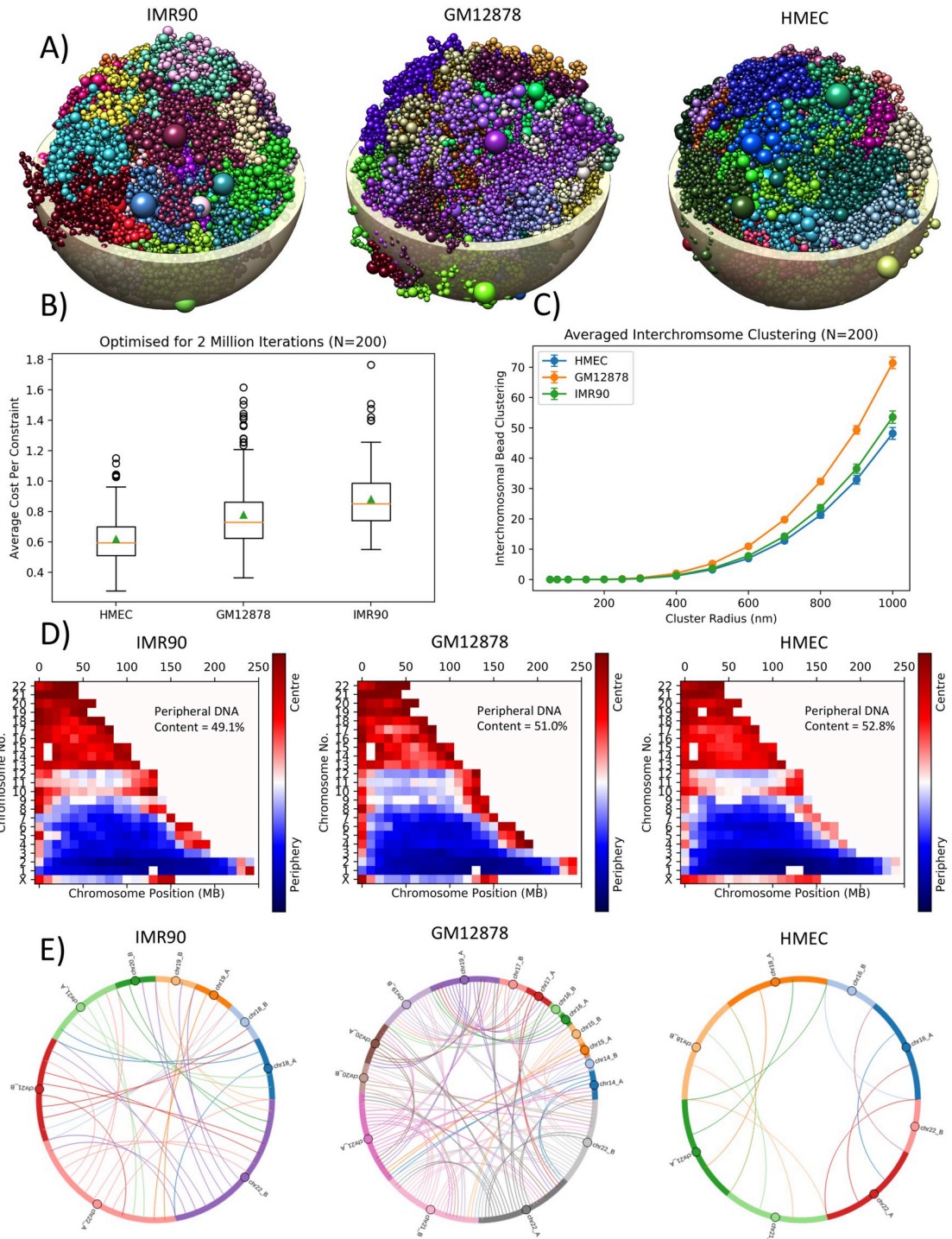

**Fig 2. Geometrical properties of the inferred 3D spatial chromatin arrangement. A)** rendered 3D example geometries for different cell-type Hi-C data. Each colour represents different chromosomes and the bead size represents the amount of DNA content. Rendered nuclear shell is to give an idea of scale for a 5 μm radius spherical nucleus. **B)** evaluation of G-NOME's ability to optimise the geometries of different Hi-C datasets. The cost function has been normalised to the total number of constraints which differs between cell types. Orange line represents the median and green triangle is the mean of the distribution. **C)** The resultant bead clustering for an increasing cluster inclusion radius. **D)** chromatin positioning within the nucleus for the different cell lines. Beads are scored based on being within the central or peripheral half of the nuclear sphere and is averaged over a 10 Mbp bin. The overall percentage of DNA content placed in the periphery is displayed within each plot. **E)** each line represents chromosomes that have interchromosomal beads within a 500 nm radius of one another. Lines are only displayed for the chromosomes that share

proximal beads in at least 50% of the examined geometries. The data shown is averaged for 200 inferred geometries for each of the cell types. Error bars are the standard deviation of 200 geometries.

geometries between interchromosomal bead clustering and cluster radius, which is not seen by the geometries formed from Hi-C data suggesting that they are not mathematically repeating arrangements (Fig 3D). This aids in the definition of an upper limit of comparison for the structured Hi-C solved geometries and shows that solved geometries contain order. To spatially analyse the geometries the normalised Ripley-K function was calculated for the 3D geometries (Fig 3E). The pseudo random geometry resulted in a straight y = 0 line for all cluster radii, suggesting it is completely spatially random. The IMR90 Ellipsoid geometry displayed an increase in clustering with a positive non-linear trend at a magnitude that was distinct to the other geometries. Whereas, the remaining geometries showed a lesser, but still non-linear separation to the y = 0 line of complete spatial randomness, suggesting some amounts of clustering, but less than the IMR90 ellipsoid geometry.

To further evaluate the spatial arrangements of the 3D geometries compared to the pseudo random geometry a proximity score was produced based on the solved 3D geometries ability

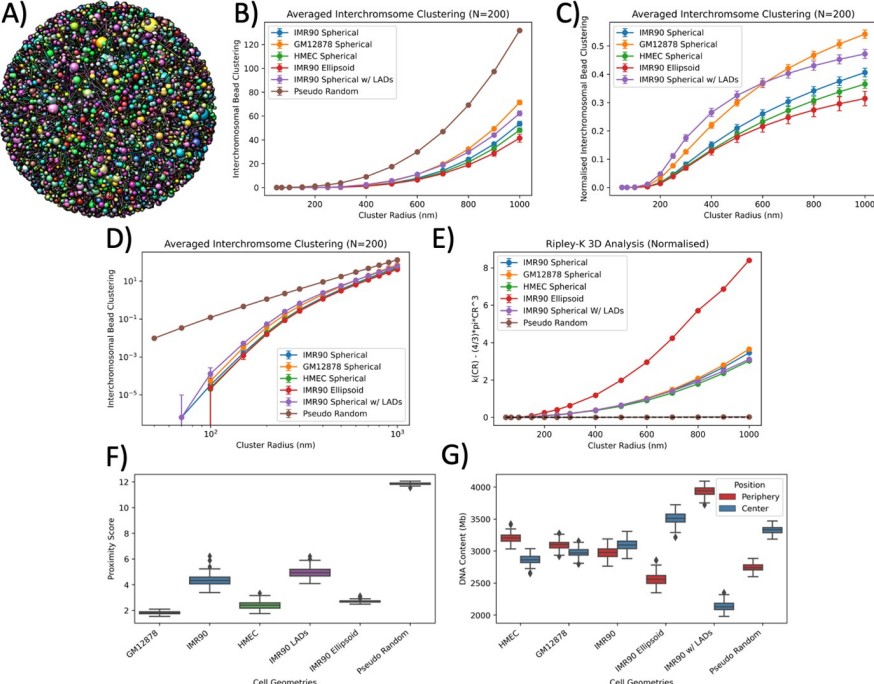

**Fig 3. Geometric comparison of solved Hi-C geometries and pseudo random geometries. A)** visualised example of a pseudo random geometry, beads are placed randomly within the restricted nuclear volume with no restriction of neighbouring beads. **B)** averaged interchromosomal bead clustering for all Hi-C cell-types and variants including the corresponding pseudo random values plotted on a linear-linear scale. **C)** normalised values of interchromosomal bead clustering for all Hi-C cell-types and variants using the corresponding pseudo random value as the normalisation parameter. **D)** same results as **B)** but on a log-log scale. **E)** Normalised 3D Ripley-K function (Eq (1)). The normalisation results in complete spatial randomness giving a y-value = 0 (shown by the dotted black line) for all cluster radii. Edge correction is applied for as $1/V_s$, where $V_s$ is the fraction of overlapping volume of the cluster radius and the nuclear radius. All values are the average of 200 inferred geometries and error bars are the standard deviation. **F)** proximity score (lower value indicates a better optimisation of the contact constraints), which is a measure of the average euclidean distance constraint placed on TADs being proximal to one another from the Hi-C data. **G)** Spatial distribution of DNA content within the inferred geometries being place in either the central of peripheral half for the nuclear volume.

to appease the Hi-C derived of TAD (bead) contact constraints. The proximity score is an averaged measure of euclidean distance for each bead with all of it's contact constraints. A lower value equivalent to a better optimisation score, for the Hi-C constraints, but ignores the constraints of all the beads being within the nucleus or lamina-based constraints. In theory this can equal zero if the analysed bead is next to all other beads it has a contact for, but given that beads have a physical size and cannot overlap this value is likely to always have some value $> 0$. The proximity score analysing (Fig 3F) shows similar levels of optimisation for all cell type and variants which are much less than the pseudo random geometry which represents a non-optimised solution. This helps justify that the optimiser is in fact promoting the constraints derived from the Hi-C data. The spatial positioning of DNA content was also evaluated (Fig 3G) to get an idea of spread in either the central or peripheral half of the nucleus volume. For geometries the three cell type geometries (GM12878, HMEC, IMR90) the distribution is relatively balanced. Wheras, the IMR90 Ellipsoid cell has increased placement of DNA content within the central portion of the cell nucleus. The IMR90 LADs have an increased amount of DNA content places in the periphery which is expected given the lamina-based constraints being applied.

## Simulated DNA damage yields in Hi-C geometries

The solved Hi-C geometries for the three cell types (GM12878, HMEC and IMR90) along with the two variants of IMR90 (IMR90-LADs and IMR90-Ellipsoid) were built in Geant4-DNA. The built Geant4 geometries were subjected to irradiation from protons, helium- and carbonions and the resultant energy depositions were classified into DNA damage (Fig 4A). The

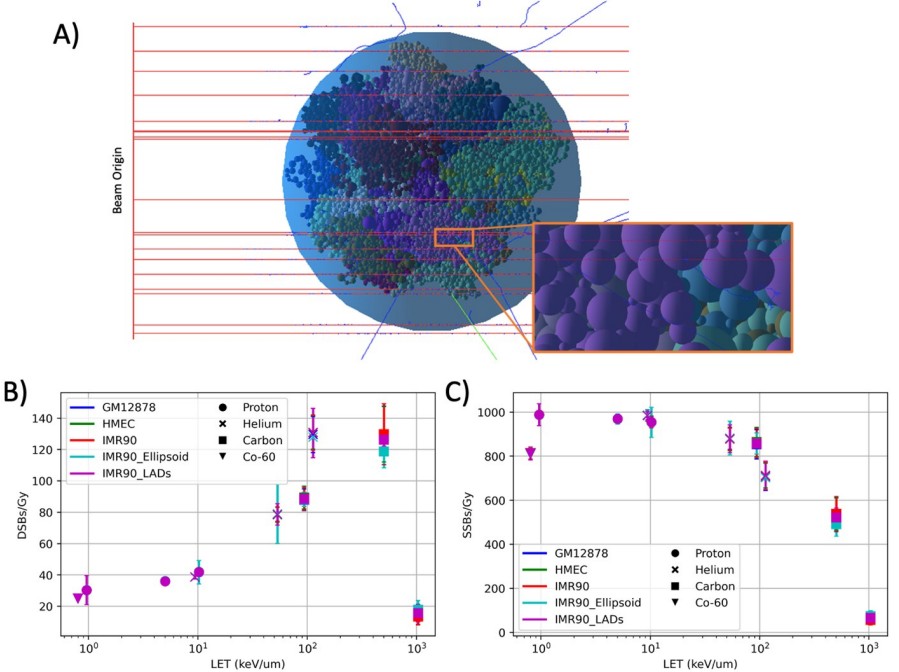

**Fig 4. Geant4 DNA damage simulation results. A)** visualisation of Hi-C geometry within Geant4-DNA being irradiated by a primary proton beam (red), generated secondary electrons (blue) and emergent gamma-rays from excitation (green). **B)** damage yields DSBs and **C)** damage yields SSBs per Gy of dose from the irradiation of Co-60 photons, protons, helium- and carbon-ions. Error bars are the standard deviation between the 200 geometries, where each geometry has 50 independent exposures.

yields of both DSB and Single-Strand Break (SSB) per unit dose (Gy) as a function of Linear-Energy Transfer (LET) (keV/$\mu$m) shows the expected increase in DSBs and a corresponding decrease in SSBs with increasing LET values (Fig 4). It can be seen that across the LET range investigated all yields remain within error bars of one another and the cell-type geometry or geometry alterations (LADs and ellipsoid shaping) do not affect resultant damage yields.

We investigated the markedly large drop in DSBs at >1000 keV/$\mu$m, which was due to increased damage clustering which results in DSBs consisting of more than two backbones, forming complex DNA damage (S1 Fig). To evaluate if there is substantial change on DNA DSB yield between different geometries of the same cell type we investigated the DSB/Gy yields as a function for each geometry (S2 Fig).

The yields per geometry were sorted (smallest to largest yields) as there was no clear correlation to specific geometries within the same cell types leading to offsets in yields. Whilst, there is some cell to cell DSB yield variation for the same cell-type it does not appear to be a significant (S3 Fig), with most points falling within the standard error of the mean from different exposures. Furthermore, to check if there were portions of the genome that were geometrically more vulnerable to DSB induction, we examined the DSB/Gy/BasePair for each chromosome, no differences were identified between cell types (S4 Fig).

To analyse how the yields of damage changes relative to the position within the cell nucleus both the DSB normalised frequency and DSB density has been plotted for each cell type (S5 Fig), which show minimal change between the cell types. The effects of adding lamina-based constraints in the IMR90-LADs geometries were evaluated against their counterpart IMR90 geometries, which do not have such constraints, in the same manner (S6 Fig). The increased positioning of beads towards the periphery does cause a notable effect for the damage distribution, with higher levels of DSB density located at the periphery.

## Simulated DNA damage distributions in Hi-C geometries

The spatial pattern of the DSBs within the cell geometry were analysed for clustering at various radii, this provided a description of the total DSB clustering (S7 Fig), the interchromosomal DSB clustering (S8 Fig) and the intrachromosomal DSB clustering (S9 Fig). These metrics can then be used to discern the intrachromosomal DSB clustering and the ratio of inter-/intra-chromosome clustering (Fig 5). Total DSB clustering, comprimised of both inter- and intra-chromosome DSBs, increases with LET, whilst the inter/intra clustering ratio has the opposite relationship with LET. This relationship is caused by the distribution of breaks making up the 1 Gy of absorbed dose within the cell nucleus. At higher LETs the distribution of damage becomes more localised, increasing the corresponding intrachromosomal DSB clustering. Conversely, at lower LETs the damage becomes more distributed, which increases the interchromosomal DSB clustering.

To evaluate the relationship between the chromosome intermingling, in the form of bead clustering (Fig 2C), and DSB interchromosomal clustering we plotted these metrics for each of the 200 geometries per cell-type category against one another (Fig 6). Whilst, there is a spread in the relationship between these two metrics it can be seen at 1 Gy of Co-60 (Fig 6A) there is a subtle increase in the mean value (denoted by the intersection of the black lines) for both geometry and damage metrics. However, as the radiation is only damaging within a small proportion of beads it was expected that the underlying relationship would strengthen as the number of damages increases (essentially sampling the geometry further). To evaluate this we also irradiated the geometries with 100 Gy of Co-60 and to show a decrease in the variation due to the exposure and a stronger observable relationship between the metrics at both a geometric and damage level. When analysing the relationship between

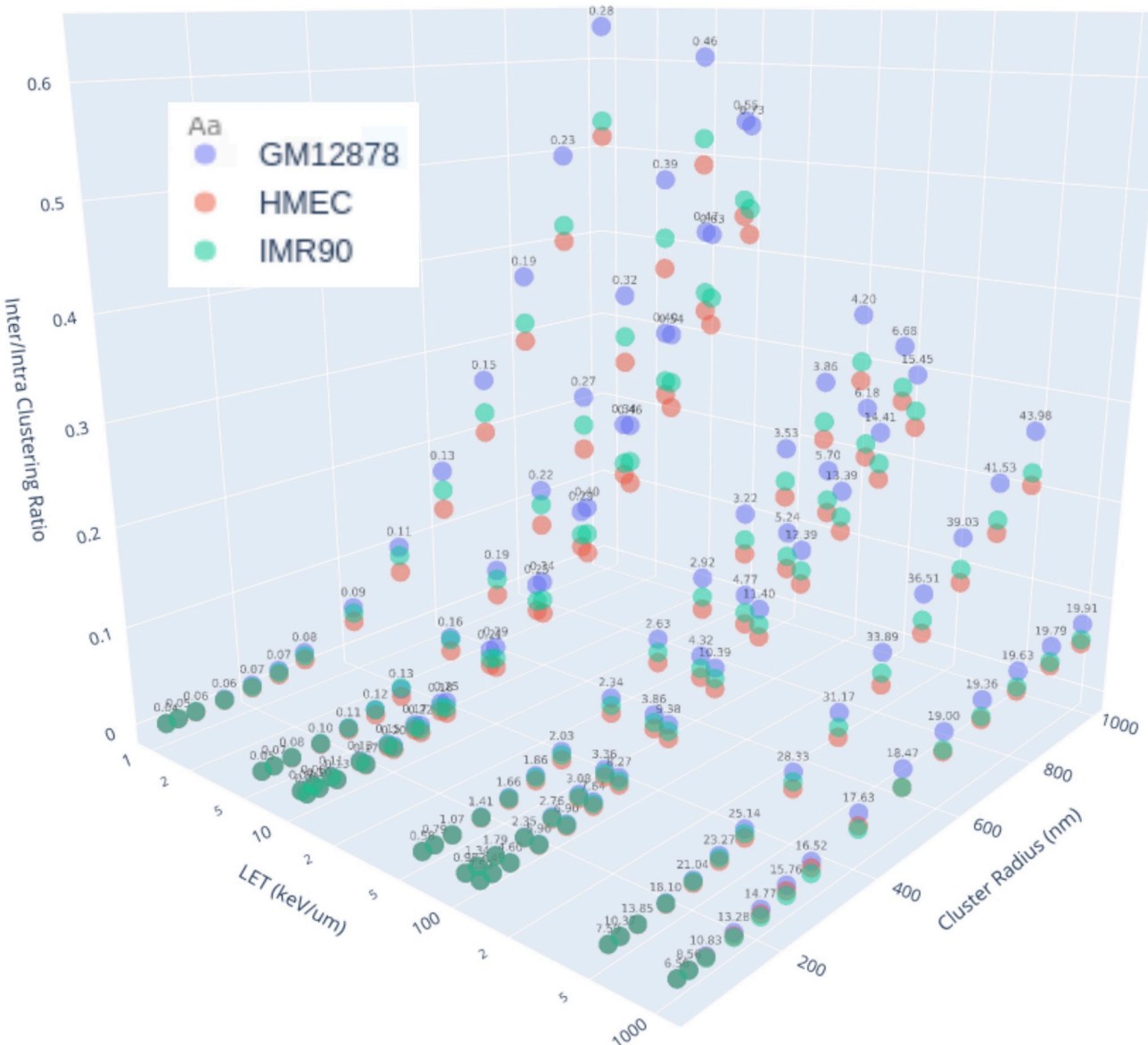

**Fig 5. Ratio of inter- and intra-chromosome DSB clustering (inter/intra CD Ratio) for a range of simulated cell-types, LET (keV/μm) and cluster radius (nm).** A lower and higher ratio indicates increased and decreased spatially clustered breaks respectively. Values of total DSB clustering are given as the floating numbers above points and are representative across cell-types. Interactive versions of this plot along with the corresponding total, inter- and intra-chromosomal DSB clustering graphs are available in S1–S8 Files. The 2D version of this plot for each fixed LET value is in S10 Fig.

geometry and DSB interchromosomal clustering for a range of LET values (Fig 6B) the relationship persists within a particular LET band (colour marked). However, with the inclusion of the two variations on the IMR90 geometry (Fig 6B-right) there is a separation from the relationship seen between cell-types (Fig 6B-left). These results identify that whilst differences between cell-types are able to be detected at the DNA damage level, the changes at this level from resultant cell shape (i.e. flattened ellipsoid cell) are noticeably different to the pattern seen across spherical shapes. The same can be said for the inclusion of LADs when solving the geometries, but on a smaller scale than that of solving for a flattened ellipsoid cell.

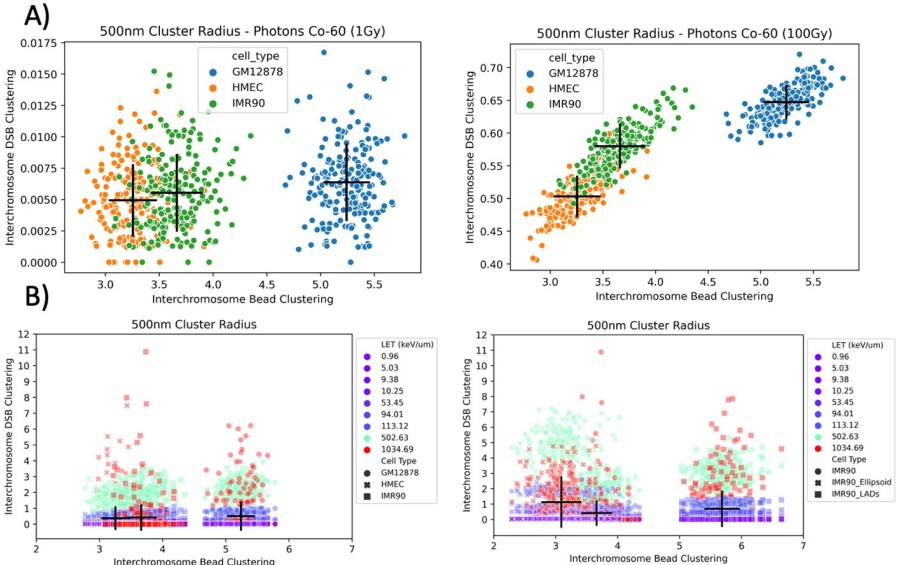

**Fig 6. Evaluation of interchromosomal geometric bead clustering and subsequent damage clustering. A)** relationship between interchromosomal bead clustering and interchromosomal DSB clustering for Co-60 photons irradiation at 1 Gy (left) and 100 Gy (right) at a 500 nm cluster radius. Black lines represent the standard deviation of each metric with the intersection corresponding to the mean value, each point represents a single G-NOME solved nucleus. **B)** same relationship of bead and DSB interchromosomal clustering, but for proton, helium- and carbon-ions of varying LET at cluster radii of 500nm with different cell-types (left) and cell variants (right).

To examine if changes in interchromosomal DSB clustering were significant each group was tested at all LET and cluster radius values shown previously (Fig 7). This identifies that changes in the interchromosomal bead clustering (Figs 1C and 2C), shown to be statistical significance (S11 Fig), translate to significant differences in the induced damage. Variation between the cell-types (Fig 7A, 7B and 7C) are typically statistically significant at higher values of LET and cluster radius (excluding carbon-ions at 1034.69 keV/$\mu$m). The largest difference when comparing cell-types is between the solved GM12878 and HMEC geometries and the smallest difference is between the IMR90 and HMEC geometries, matching the differences observed in (Fig 6). Whereas, statistically significant differences can be seen for the majority of tested LETs and cluster radii in the variant comparisons (Fig 7D, 7E and 7F).

## Discussion

We have shown that the changing of cell-type, addition of LADs, or solving for an ellipsoid does not have noticeable influences on the yields of DSB and SSB DNA damage, but there is a difference in the pattern of damage. The resultant DNA DSB interchromosomal clustering (Fig 6B) suggests that it is possible to detect statistically significant differences between cell types (Fig 7), but the addition of LADs and especially ellipsoid shaping causes a distinct alteration in how the interchromosomal geometric description propagates through to the damage distribution. It is in this case that the normalised Ripley-K may be a better predictor of the DSB interchromosomal clustering when you depart from spherical geometries as this accounts for non-overlapping cluster radii volume and nucleus volume through boundary corrections. These variations between cell-types are detectable in all types of clustering (interchromosomal, intrachromosomal and total), with the magnitude of the values still dictated by radiation parameters (e.g. LET). The observed changes in inter/intra chromosomal DSB clustering ratios is an interesting parameter to analyse as it may be a predictor of inter- and intra-chromosomal

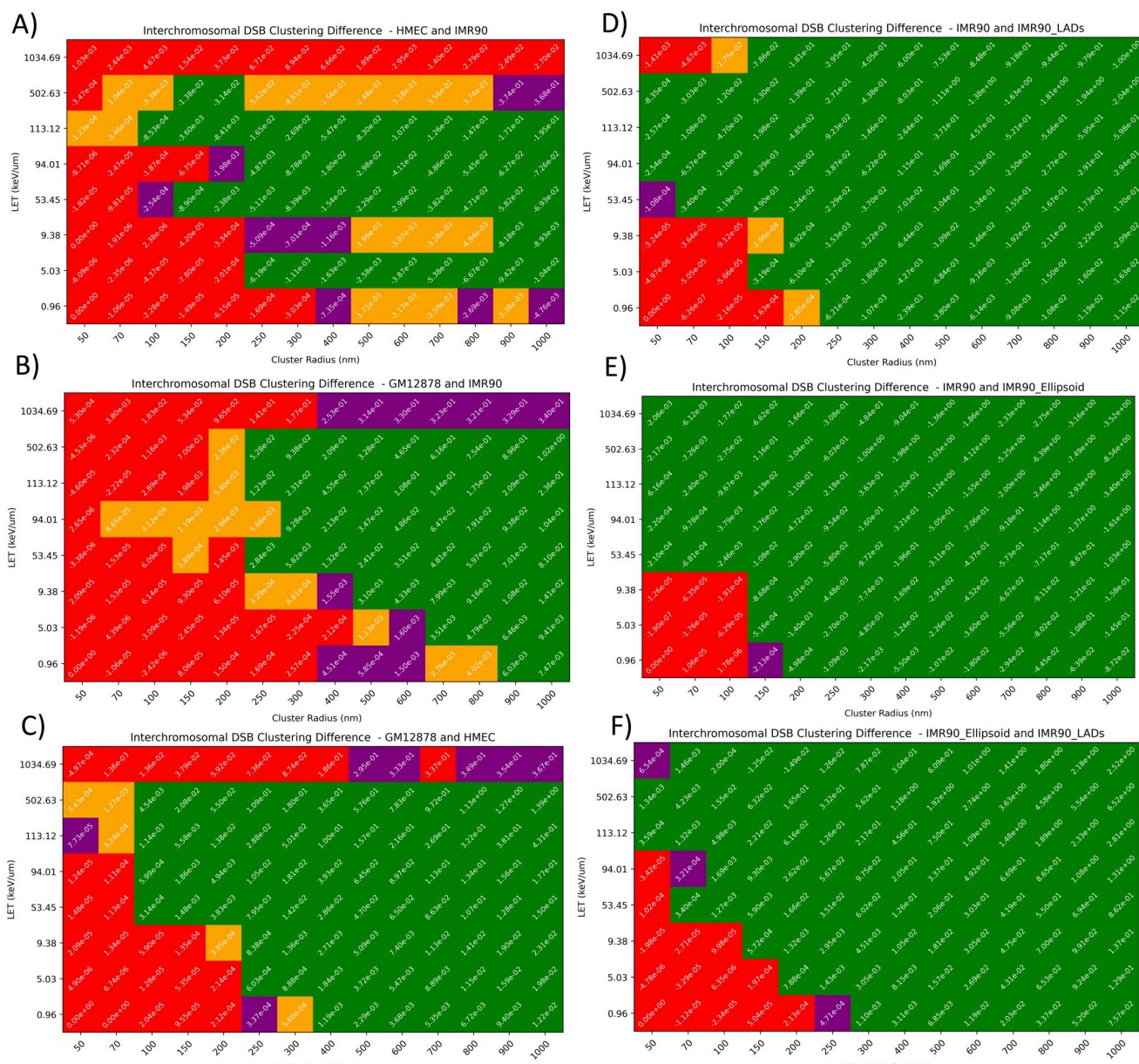

**Fig 7. Examination of statistically significant change in interchromosomal DSB clustering between cell-types (A—C)) and variants (D)—F)).** The values of the heat maps is the difference in the average interchromosomal clustering between the two cell types. The heat maps colouring shows the false discovery rate adjusted P-values from the Kolmogorov–Smirnov two-sided test for the full range of LET (keV/$mu$m) and cluster radius (nm). The null hypothesis of the test is that two independent samples are drawn from the same continuous distribution. Colour coding for adjusted P-values at varying thresholds: red (P > 0.05), purple (0.05 > P > 0.01), yellow (0.01 > P > 0.001) and green (P < 0.001). In this case, distributions with adjusted P-values < 0.05 are considered as statistically significantly different. Each of the tested distributions had 200 geometries per cell-type or variant group. Each value of interchromosomal DSB cluster density is an averaged result which comes from 50 independent 1Gy exposures at the listed LET and cluster radius values.

misrepair, which has a differing biological response [46]. We believe that interchromosomal DSB clustering may be a good predictor of interchromosomal chromosome aberrations, which are often toxic to the cell [47]. Experimental work has previously shown increased chromosomal intermingling correlates with increased chromosomal translocation [15], we are able to explore these findings at their intermediate step of DNA damage distributions.

To resolve if it is feasible to detect differences at a DNA damage level when using different Hi-C cell-type datasets we ensured that all geometries were solved for the same nuclear volume and the total amount of DNA (6 Gbp). However, from the literature, it is apparent that cell nucleus [48] size can vary substantially both between and within cell-type. This change in cell size has also been linked to a change in the amount of DNA content [49], these features have not been encapsulated here due to simulation time, but are within the capability of the G-NOME software and our damage model. Due to the differences noted from changes in cell shape, we would expect the differences in both size and DNA content to have a marked response in the damage distribution produced. Furthermore, through changing the amount of DNA content there is an alteration in the number of sensitive targets within the cell, which would be expected to lead to a difference in the yields of damage.

In this study, we have included the effects of solving the geometry of the same volume, but for a flattened ellipsoid shape (which may be predominant *in vitro*) and comparing it to the corresponding spherical geometry solved using the same Hi-C data. This was perfomed to understand any radiation damage differences that may occur in a flattened nucleus, a phenomenon that is observed during cell spreading when cells are plated in a 2D system [50]. As it is common to plate cells in some irradiation experiments the substantial change in the damage distribution within the nucleus may indicate differences in the expected radio-response between 2D and 3D cell system experimental techniques. The possibility to evaluate the geometric effects of nuclear shaping in attached 2D versus 3D cell systems may help discern experimental variability observed across different techniques analysing the same endpoint [51]. There are further implications of this when we move towards modelling patient radio-response, previous work has shown that there is improved predictive power of clinical efficacy when using 3D over 2D *in vitro* systems [52]. This is clearly an issue as a large quantity of previous and continuing experimental radiobiological work are carried out in 2D cell systems, here we are able to understand the differences in radio-response due to nuclear shape change and open the possibility of translation factors to be derived.

The geometries produced are representative of a single cell but are derived from a highly averaged population dataset. By enforcing the polymer model to arrange itself through a series of movements we are able to use the Hi-C data as a guide to achieving realistic single-cell conformations, rather than positioning beads with no heuristic limitation to achieve the smallest cost function. With the increased development in single-cell Hi-C [53] a development of the G-NOME software would be to use data to perform single-cell conformation modelling. This would allow a method of validation for using the current population-based method as a suitable approximation for single-cell conformations. However, the benefits of single-cell Hi-C extend beyond simple validation, it may allow for the ability to describe different cell sub-types, which includes cell-cycle specific states [20, 54]. The ability to discern cell-cycle specific geometries is a pertinent goal in radiobiological modelling, as there is a well establish variation in radiosensitivity at different cell cycle phases [55]. Leveraging differences in chromatin conformation along with alterations in the active repair pathways could be a viable method of modelling these effects.

The current model is limited in terms of its genomic resolution, at present the size of each bead is defined by topologically associated domains (TADs), which are used widely as discerning DNA segments of self-interacting regions. There may be a benefit in trying to build models at smaller genomic resolutions, but there should be caution in the decreased signal-to-noise ratio and the possibility of mis-classifying significant contact points. Although, as the field of Hi-C moves to increasingly finer resolutions, including the recent inception of Micro-C [56], there may be opportunity to model sub-structures (e.g. nucleosome) of the genome. These sub-structures may be especially important when attempting to model the more granular

configuration of the DNA damage, along with the ability to model regions of heterochromatin and euchromatin.

In order to accurately model differences in intrinsic radio-sensitivity between cell-types we must evaluate the the characteristic differences, starting with DNA damage. We have shown that there is differences in DNA damage that are geometry-driven and believe this is key to understanding the effects of LET on relative biological effectiveness through the explicit modelling of chromosome aberration probability. Furthermore, through examination of the effects of LET and cluster radius leading to significant differences in simulated interchromosomal DSB clustering of different cell types (Fig 7), it is possible to use this information to design experiments to evaluate DSB end motion, a subject that will drive chromosome aberration probability, but remains elusive with proposed DSB end motion being in the range of 70 [57], 100 [58] and 500 nm [59]. The related topic of chromatin dynamics following irradiation, that has recently been shown to be cell-type specific [60], will also need to be examined to predict the downstream biological effects from the cell-type specific damage patterns. Finally, we acknowledge that to fully encapsulate the radiobiological differences between cell-types we must include descriptions of the characteristic variation that are omics-driven [61].

The combination of Hi-C data to inform geometric structures for *in silico* modelling of radiation track structure and DNA damage has been shown to be feasible in this study. Through leveraging the genomic element of the Hi-C data it is possible to enrich the descriptions of radiation damage and quantify the overall damage distribution. This will be used in subsequent models of DNA repair, where cell-type differences of proliferation rate, protein expression and micro-environment can be encapsulated to further investigate the observed variation in cellular radio-response. Furthermore, through this expansion into Hi-C modelling for normal cells in an attempt to better understand normal tissue response, we can look towards incorporating the alterations in genomic structure found in cancer cells to improve our understanding of target response [62–64]. It is thought that this relationship will work symbiotically, as radiobiological response is strongly dependent on geometry (e.g. translocations) [15] and can be used as an additional experimental technique to validate the Hi-C geometries. Finally, though the use of radiobiological modelling there forms a new translational pathway to how the improved geometric understanding of Hi-C could benefit cancer patients at a clinical level.

## Materials and methods

### Preparation of Hi-C data

This introduced model has been designed to utilise the same input gtrack file format as previous 3D inference model "Chrom3D", enabling previous protocol work [65] that outlines how to analyse and process gtrack files to still be applicable for the G-NOME software. Furthermore, this also enables the ability to incorporate lamina-associated domains (LADs) data into the optimisation objectives to promote areas of the genome which have higher interaction with the nuclear lamina to the periphery of the solved geometry [66, 67].

Whilst the previous protocol work [65] should be referred to for a detailed explanation of how to process a Hi-C dataset to get a gtrack file for subsequent G-NOME solving, we provide a short overview here. The gtrack file is a line by line summary of the TADs identified within the Hi-C data, it includes the chromosome identity, genomic start position, genomic end position, unique ID, contacts and optional periphery status (binary). To generate this information tools such as aidenlab's juicer (https://github.com/aidenlab/juicer/) should be used to pre-process the raw contact data into *.hic files. After which the Arrowhead algorithm should be used to deduce contact domains and will provide the starting files for the protocol outlined by

Paulsen et al [65]. These domains can then be examined at a intrachromosomal and interchromosomal level, to establish TAD interaction counts ready for analysis and resulting in the creation of a BEDPE-format file. This process can take significant time and will depend on the resolution of the Hi-C matrix. It is at this point to account for domains with centromeres and unmappable regions to avoid artifacts in the statistical tests. The P-values and odds ratios are calculated for all the contacts detailed in the BEDPE files using the non-central hypergeometric (NCHG). The data is then filtered using both the calculated false discovery rate (FDR) and the odds ratio. It is at this point that the data can be formatted in the gtrack file format, along with the chance to aggregate corresponding Lamina-associated domain (LAD) data if required.

When using the protocol for generating the gtrack files, we ensured that variation in the analysed intra-chromosomal (down to 5 kbp) and inter-chromosomal (100 kbp—1 Mbp) interaction matrix gave no discernible difference in the geometric analysis when solving the geometry at a TAD resolution (S12 Fig). All cell-type and variant geometries shown in the results have been solved using the 50 kbp intra-chromosomal and 1 Mbp inter-chromsomal matrix for the production of the gtrack file.

This study uses the Hi-C data published by Rao et al., [68] (GEO Accession GSE63525), which has been used widely within the literature as an exemplar test dataset. However, the proposed model is suitable for inferring 3D geometries from other Hi-C datasets.

## Hi-C solver

The G-NOME software uses a Markov-Chain Monte Carlo (MCMC) polymer model, which can be optimised through both the metropolis-hastings and simulated annealing algorithms. The resultant geometry is based on a series of optimised objectives applied to polymer beads, which represent a defined number of base pairs of DNA along a chromosome (TADs), being arranged so that areas of strong contact probability are spatially proximal. In order to arrange the geometry a series of iterations attempt several bead movement types at random, if the movement results in an improved objective score the iteration can be accepted and the next iteration may take place. Running through many iterations allows for the overall solved geometry to be realised. The movement types are as follows:

1. Crankshaft—rotates a random number of beads around two fixed beads points.

2. Arm Rotation—from a random position along the chromosome rotate all beads to the end or start of the the chromosome. The start or end is chosen by random.

3. Arm Wiggle—from a random position along the chromosome re-position (via a self avoiding walk) beads to the end or start of the chromosome. The start or end is chosen by random.

4. Translation—move the whole chromosome by a random x, y, z between 0 and 1um.

5. Rotation—rotate the whole chromosome by a random amount.

The G-NOME software developed is a re-implementation of the Chrom3D model [23], but includes modifications to better interface with track-structure models of DNA damage [57, 69]. The resultant 3D structures from G-NOME have been compared with Chrom3D for DNA-content spatial placement (S13 Fig) and the optimisation score it delivers (S14 Fig) to ensure no major deviation between the models given for the same set-up. This improved interfacing allows for geometric solving and DNA damage models to be coupled, resulting in generation of unique cell geometries for every run of the DNA damage model. Further differences between the introduced G-NOME software and Chrom3D include: change in the

programming language to Python, improved compute efficiency resulting in faster solving times (S15 Fig), the ability to optimise for non-spherical geometries (here we demonstrate ellipsoid) and custom optimisation routines. Custom optimisation routines refers to the ability to be able to dynamically alter optimisation constraints during a run time session. We believe this gives us additional flexibility that may be required in future work with G-NOME and radiobiological modelling.

As Hi-C data is the averaged result of a large cell population it becomes apparent that the solution space for solving these geometries may contain many suitable solutions of chromosome conformation. This has been shown in several studies [23, 24] and is the underpinning for the hypothesis of encapsulating the biological variation by using different solutions outputted from the model. Variation in the output arises from the Monte Carlo approach of using different random seeding for the initial distribution of the chromosomes, since the initial distribution has a strong influence on the end result.

In total 1,000 geometries (200 geometries per group) have been calculated which include: three cell types (IMR90—human fetal lung fibroblast, HMEC—human adult mammary epithelial and GM12878—human B-lymphocyte), two cell shapes (spherical and ellipsoid) and inclusion of LADs for the IMR90 cell line. All spherical cells were solved for the same target nucleus size of 5 $\mu$m radius. The ellipsoid cells were solved for the target nucleus size of 1.0x11.8x11.8$\mu$m radii. All cells were modelled with the same amount of DNA content (6Gbp) and as normal diploid human cells (46 chromosomes). All geometries have been optimised with 2 million iterations of successful movements unless optimised solutions were found before this or stated otherwise. All renderings of the 3D spatial chromatin arrangement were made using the 3D visualisation tool Chimera [70] by loading the "*.cmm" files created from the G-NOME software. All geometries were solved using additional nuclear boundary constraints, which adds a cost based on if the beads were confined to the user-defined nuclear boundary. The costs applied to these constraints are 0 if the bead is within the cell nucleus and only occur cost on positioning outside of the nucleus based on the euclidean distance from the nuclear boundary. This constraint can be toggled when using the simulation run script provided through the flag "–ConstrainNucleus". These additional constraints were required as we wanted to preserve total volume across all cell geometries for the subsequent Geant4 simulation.

## DNA damage simulation

Details of the polymer beads, produced by G-NOME, are read into our DNA damage application [57, 69]. Each bead is placed as a spherical geometry object in the Monte Carlo toolkit Geant4 (geant4 10.5.1) [71], using the X, Y, Z and variable bead radius, within a bounding nucleus volume. Simulation of cell nucleus irradiation is performed within Geant4, using the default Geant4-DNA physics list [44]. Within Geant4 the track structure of a radiation source is simulated as a series of interaction limited steps through a specified geometry, with each step updating the energy and trajectory of the primary or secondary particle. For particle-induced DNA damage, energy depositions occurring within beads are recorded. Two conditions are applied to convert energy depositions into strand breaks. Firstly, a spatial sampling of 14.1% is applied to the bead. Secondly, an energy range probability is applied, from 0 at 5 eV to 1 at 37.5 eV. Once passing both conditions an energy deposition is accepted as a strand break and is randomly assigned to strand 1 or 2 of the double helix, with equal probability. The chromosome of damage is directly assigned from the G-NOME bead. The position along the chromosome is informed by the G-NOME bead. Since each bead contains a portion of the chromosome a minimum and maximum base pair position is known, with the actual position

of damage taken as a random point between the two. Following the simulation of all particles required to deliver a dose the list of damages is analysed through a clustering algorithm. Double Strand Breaks are formed by two or more strand breaks that are on opposite strands and separated by 3.2 nm or less (equivalent to 10 bp). Strand breaks that don't form a DSB are classified as Single-Strand Breaks.

The spatial sampling of 14.1% corresponds to a sensitive fraction of the bead and was determined to reproduce DSB yields seen in our previous work [57]. This fit is in good agreement with other models [72, 73] which have been validated against experimental data. The energy-based probability of damage induction was adapted from PARTRAC [73] and is based on studies of DNA strand breakage after exposure to very low energy electrons [74].

For photon induced DNA damage, DSB induction is assumed to follow a Poisson distribution with an average of 25 DSBs/Gy. For each DSB the chromosome is chosen at random, with probability weighted according to the chromosome size relative to total genome size. Similarly, a bead within that chromosome is chosen randomly, with probability weighted according to on bead size relative to the sum of all beads forming the chromosome. A random X, Y, Z within the selected is bead is assigned to the DSB. All DNA damages are recorded in the Standard DNA Damage (SDD) format [40].

In this work, the G-NOME cell models are irradiated with 1 Gy of photons, protons (3 MeV—67 MeV), helium ions (4 MeV—80 MeV), or carbon ions (10 MeV—213 MeV). Ensuring a range of particle type and track-averaged Linear Energy Transfer, with an overlap in LET between the particles. A table of all particles, particle energies and calculated LET values are given in the S1 Table. A reader for these geometries will be implemented in "TOPAS-nbio" to further accessibility.

## Statistical information

The statistical analysis used in Fig 7 was an adjusted P-value using the using the Benjamini-Hochberg correction to control the false discovery rate (type I error). The P-values calculated for adjustment are from a Kolmogorov–Smirnov test between different cell-type or variant groups. The test was two-sided with the null hypothesis indicating that the two samples are drawn from the same continuous distribution. The adjusted P-values was presented for all cell-types and variants groups for a range of track averaged LET values and cluster radii. A threshold of $P < 0.05$ was used to determine statistically significant differences between tested samples. The Kolmogorov–Smirnov test was calculated using Python and the scipy (v1.4.1) package. The Benjamini-Hochberg correction was calculated using Python and the statsmodels (v0.12.0) package. All tests had 200 geometries in each group.

The Ripley-K function was calculated for the 3D distrubtion of polymer beads for each 3D geometry produced by G-NOME. The calculation methodology follows the 3D implementation carried out by Jafari-Mamaghani et al [75]. The equation is detailed below:

$$K(CR) = V_{Nuc} \frac{\sum_{i=1}^{n} \sum_{i \neq j} I[D(i,j) \leq CR]}{V_s n^2} \tag{1}$$

where $V_{Nuc}$ is the volume of the nucleus, CR is the cluster radius, n is the number of DSBs in the nucleus, $V_s$ the edge correction term is the fraction of overlapping volume of the CR volume and the $V_{Nuc}$ and I is the indicator function which will be either 1 if the condition $D(i,j) \leq t$ is true or will be 0, $D(i,j)$ is the euclidean distance from DSB i to DSB j.

## Supporting information

**S1 Fig. DSB damage complexity.** Average number of DSBs/Gy (filled symbols) and back-bones per DSBs (empty symbols) for a range of LET values across different particle types. Error bars are displayed as the standard error of the mean for 100 repeats.
(TIF)

**S2 Fig. DSB yield variation across generated geometries.** Yields of DSB per Gy of dose for the each of the 200 geometries created. Results have been sorted from smallest to largest yields to allow for easier interpretation. Different cell-types are shown as different line types with each radiation quality presented as a different colour. Errors are the transparent area around the line and are the standard error in the mean for 50 independent exposures per geometry.
(TIF)

**S3 Fig. DSB yield distribution.** Double-strand break yield histograms for 200 geometries of each cell-type and variant.
(TIF)

**S4 Fig. DSB yield chromosome distribution.** Double-strand break per 1Gy of dose per DNA basepair on each of the modelled 46 chromosomes for each cell-type and variant. Error bars are displayed as the standard error of the mean for 200 geometries for each cell-type and variant with each geometry having 50 independent exposures.
(TIF)

**S5 Fig. Spatial distribution of DNA DSB yields for different cell types.** Dual axis plot—left y-axis shows the histogram plot of the Normalised DSB frequency and right y-axis is the corresponding average DSB density for the same x-axis bin per geometry. Both are given as a function of distance from the nucleus centre. The cell types are all solved for a spherical nucleus and do not include LADs. The DSB frequency was normalised to the maximum number of DSBs within any bin for a given cell type. DSB density is calculated as the average number of DSBs per geometry (N = 200) within a bin divided by the volume ($\mu m^3$) of the spherical shell of the bin. Error bars in the DSB density are the standard error in the mean for all 200 geometries for each cell type.
(TIF)

**S6 Fig. Spatial distribution of DNA DSB yields for the addition of LADs.** Dual axis plot—left y-axis shows the histogram plot of the Normalised DSB frequency and right y-axis is the corresponding average DSB density for the same x-axis bin per exposure. Both are given as a function of distance from the nucleus centre. Comparison between IMR90 with and without LADs constraints for a spherical nucleus. The DSB frequency was normalised to the maximum number of DSBs within any bin for a given cell variant. DSB density is calculated as the average number of DSBs per geometry (N = 200) within a bin divided by the volume ($\mu m^3$) of the spherical shell of the bin. Error bars in the DSB density are the standard error in the mean for all 200 geometries for each cell variant.
(TIF)

**S7 Fig. DSB clustering per Radiation Quality Plots.** Double-strand break clustering as a function of the cluster radius for all cell-types and variants. Error bars are displayed as the standard error of the mean for 200 geometries for each cell-type and variant with each geometry having 50 independent exposures.
(TIF)

**S8 Fig. DSB Interchromosomal clustering per Radiation Quality Plots.** Double-strand break interchromosomal clustering as a function of the cluster radius for all cell-types and variants. Error bars are displayed as the standard error of the mean for 200 geometries for each cell-type and variant with each geometry having 50 independent exposures.
(TIF)

**S9 Fig. DSB Intrachromosomal clustering per Radiation Quality Plots.** Double-strand break intrachromosomal clustering as a function of the cluster radius for all cell-types and variants. Error bars are displayed as the standard error of the mean for 200 geometries for each cell-type and variant with each geometry having 50 independent exposures.
(TIF)

**S10 Fig. DSB inter/intra clustering per Radiation Quality Plots.** Double-strand break inter/intra chromosomal clustering as a function of the cluster radius for all cell-types and variants.
(TIF)

**S11 Fig. Statistical differences between interchromsomal bead clustering of different cell lines.** False discovery rate adjusted P-values from a 2-sided Kolmogorov-Smirnov test on the interchromosomal bead clustering values for the different cell-types and variants. Colour coding for adjusted P-values at varying thresholds: red ($P > 0.05$), purple ($0.05 > P > 0.01$), yellow ($0.01 > P > 0.001$) and green ($P < 0.001$). In this case, distributions with adjusted P-values $< 0.05$ will be considered as having significant statistical difference to one another. Each of the tested distributions had 200 geometries per cell-type or variant group.
(TIF)

**S12 Fig. Nuclear positioning of beads at varying Hi-C contact resolutions.** Bead positioning between periphery and central locations for a range of different interchromosomal contact resolutions at the finest available intrachromosomal contact resolution. Each category consists of 200 geometries created from the corresponding gtrack file created from using different analysis resolutions.
(TIF)

**S13 Fig. DNA content position model comparison.** Box plots of the DNA content positioned either in the peripheral half or central half of the cell nucleus volume. These results are for 50 geometries from G-NOME and 50 geometries from Chrom3D (v1.0.2). In both models the same input IMR90 noLADs gtrack file was optimised for 1 million iterations, 5-micron nuclear radius and 0.15 occupancy volume.
(TIF)

**S14 Fig. Proximity score model comparison.** Box plots of the proximity scores which is the average Euclidean distance between TADs which have a constraint to be proximal other TADs (lower value indicates a better optimisation of the contact constraints). To put these differences into perspective for a randomly distributed geometry where the proximity score is approximately 12. These results are for 50 geometries from G-NOME and 50 geometries from Chrom3D (v1.0.2). In both models the same input IMR90 noLADs gtrack file was optimised for 1 million iterations, 5-micron nuclear radius and 0.15 occupancy volume.
(TIF)

**S15 Fig. Nominal evaluation of speed performance.** Timing performance for a nominal single IMR90 spherical cell 3D geometry generation using both G-NOME and Chrom3D (v1.0.2). In this case for 2 million iterations (the number used for the evaluation of different

cell types) is 23.5 hours in Chrom3D and 7.6 hours in G-NOME.
(TIF)

**S1 Table. Incident particle range.** Examination of particle range travelling through $20\mu$m of water. Simulation was carried out for each energy with the furthest depth of each particle scored and averaged for 100 single-particle transversals. Particles which transverse beyond the $20\mu$m of water are simply signified as having ranges beyond $20\mu$m.
(TIF)

**S1 File. Cell type interchromosomal clustering.** Interactive 3D plot of interchromosomal DSB clustering for a range of cluster radii and LET for the three cell types: IMR90, GM12878 and HMEC.
(HTML)

**S2 File. Cell type intrachromosomal clustering.** Interactive 3D plot of intrachromosomal DSB clustering for a range of cluster radii and LET for the three cell types: IMR90, GM12878 and HMEC.
(HTML)

**S3 File. Cell type inter/intra ratio clustering.** Interactive 3D plot of the ratio of interchromosomal/intrachromosomal DSB clustering for a range of cluster radii and LET for the three cell types: IMR90, GM12878 and HMEC.
(HTML)

**S4 File. Cell type clustering.** Interactive 3D plot of the total DSB clustering for a range of cluster radii and LET for the three cell types: IMR90, GM12878 and HMEC.
(HTML)

**S5 File. Cell variation interchromosomal clustering.** Interactive 3D plot of interchromosomal DSB clustering for a range of cluster radii and LET for the three IMR90 cell variants: IMR90, IMR90 with LADs and IMR90 ellipsoid.
(HTML)

**S6 File. Cell variation intrachromosomal clustering.** Interactive 3D plot of intrachromosomal DSB clustering for a range of cluster radii and LET for the three IMR90 cell variants: IMR90, IMR90 with LADs and IMR90 ellipsoid.
(HTML)

**S7 File. Cell variation inter/intra ratio clustering.** Interactive 3D plot of the ratio of interchromosomal/intrachromosomal DSB clustering for a range of cluster radii and LET for the three IMR90 cell variants: IMR90, IMR90 with LADs and IMR90 ellipsoid.
(HTML)

**S8 File. Cell variation clustering.** Interactive 3D plot of total DSB clustering for a range of cluster radii and LET for the three IMR90 cell variants: IMR90, IMR90 with LADs and IMR90 ellipsoid.
(HTML)

## Acknowledgments

This project utilised a significant amount of computational resources (approx. 120 CPU/years). The authors would like to acknowledge the significant amount of assistance given by University of Manchester Research IT, with special appreciation to Dr. Daniel Corbett. The computational work was carried out with the use of both the Computational Shared Facility

and Condor at The University of Manchester. Molecular graphics and analyses performed with UCSF Chimera, developed by the Resource for Biocomputing, Visualization, and Informatics at the University of California, San Francisco.

## Author Contributions

**Conceptualization:** Samuel P. Ingram, Nicholas T. Henthorn, John W. Warmenhoven, Norman F. Kirkby, Ranald I. Mackay, Michael J. Merchant.

**Data curation:** Samuel P. Ingram, Nicholas T. Henthorn.

**Formal analysis:** Samuel P. Ingram, Nicholas T. Henthorn.

**Funding acquisition:** Karen J. Kirkby.

**Investigation:** Nicholas T. Henthorn, John W. Warmenhoven, Norman F. Kirkby, Ranald I. Mackay, Michael J. Merchant.

**Methodology:** Samuel P. Ingram, John W. Warmenhoven.

**Project administration:** Karen J. Kirkby, Michael J. Merchant.

**Resources:** Michael J. Merchant.

**Software:** Samuel P. Ingram, Nicholas T. Henthorn.

**Supervision:** Norman F. Kirkby, Ranald I. Mackay, Karen J. Kirkby, Michael J. Merchant.

**Validation:** Nicholas T. Henthorn, Norman F. Kirkby, Karen J. Kirkby.

**Visualization:** Samuel P. Ingram.

**Writing – original draft:** Samuel P. Ingram.

**Writing – review & editing:** Samuel P. Ingram, Nicholas T. Henthorn, John W. Warmenhoven, Michael J. Merchant.

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
