## [Decision Letter · Decision Letter 0]

17 Aug 2020

Dear Mr Ingram,

Thank you very much for submitting your manuscript "Hi-C implementation of genome structure for in silico models of radiation-induced DNA damage" for consideration at PLOS Computational Biology.

As with all papers reviewed by the journal, your manuscript was reviewed by members of the editorial board and by several independent reviewers. In light of the reviews (below this email), we would like to invite the resubmission of a significantly-revised version that takes into account the reviewers' comments.

We cannot make any decision about publication until we have seen the revised manuscript and your response to the reviewers' comments. Your revised manuscript is also likely to be sent to reviewers for further evaluation.

Sincerely,

Ferhat Ay, Ph.D

Associate Editor

PLOS Computational Biology

William Noble

Deputy Editor

PLOS Computational Biology

Reviewer's Responses to Questions

**Comments to the Authors:**

Reviewer #1: Ingram et al., in this manuscript, integrate genome organization information from Hi-C data and radiation track structure modeling to understand DNA damage within cell-line specific genomes. Understanding the radiation-induced DNA damage has a critical relevance in different fields of research, especially cancer therapy or radiotherapy. The correct radiation dosage is extremely important to minimize the side effects of radiotherapy on healthy tissues while maximizing the effects on cancerous ones. Although a general safe amount of dosage is known, much improvement and understanding are still required in terms of the way the DNA is damaged to overcome the worst side-effects. The authors hypothesize that genome organizational features partially drive radiation effects on different cell types. Ingram et al., here coupled Hi-C inferred 3D geometries from three different cell lines (IMR90, HMEC, and GM12878) with radiation track modeling to understand the cell line specific radiosensitivity and the simulation results do show some pattern of DNA damage when exposed to radiation. I think the research work performed in this manuscript is important and adds clinical relevance to the utilization of Hi-C data. However, after going through the body of work I have the following major concerns -

1. Although the focus of the manuscript is not on the 3D genome modeling and its validation, the results heavily depend on the exact fact. The authors in the manuscript re-implemented the Chrom3D program as G-NOME to model the 3D genome organization from Hi-C data. They did mention some differences between the G-NOME and Chrom3D programs but without any comparative result. It is not clear to me about the novelty of re-implementing the original algorithm in some other programing language without any clear evidence of benefit. The modeling of ellipsoid geometry seems interesting as it is a novel feature highlighted by the authors to the G-NOME program, but the authors didn't provide any information on how they generated ellipsoid models and their biological validity as compared to spherical ones. Figure 1d shows the chromatin position of beads from IMR90-spherical, w/LADs, and IMR90-ellipsoid model. IMR90-ellipsoid, as compared to the spherical model shows very few beads near the nuclear periphery, is this only an optimization issue? I would suggest the authors correlate the model with LAD and gene-expression information to validate the ellipsoid model and the overall improvement of the G-NOME program. What are the "custom optimization" routines in G-NOME? And how they affect 3D modeling? Where are the generated 3D models?

2. I am not clear about the "two conditions" and assignments of the DSB, SSB described in the DNA simulation section. What is the "spatial sampling of 14.1%" means as the first condition? And how come the authors arrived at this exact number? In the second condition, probabilities corresponding to energies ranging from 5 to 37.5 eV are assigned. Is this a standard range of energies during simulation, and where are the references supporting the procedure? The author's talks about energy values in KeV and MeV range, how are these high energies converted to probability space? The 3D geometries are solved from a low-resolution Hi-C matrix, but how these low-resolution beads are used to approximate double or single-strand DNA breaks is not clear to me. The authors mention that DSB induction is assumed to follow a Poisson distribution, and chosen randomly with probability weighted according to the chromosome and bead size. The chromosome size remains invariable across cell types, and only the bead size at TAD resolution should provide some cell line specific weightage to determine the probability for DSB induction. But the 3D modeling of a genome from Hi-C data provides the spatial position of a bead relative to the nuclear lamina (Figure 1d). From previous studies, it is known that this relative position (periphery to central) is highly variable across genomes and should provide the most cell line specific information. The function to weigh this spatial positioning of a bead relative to the nuclear periphery and energy source while calculating the probability to induce DSB is missing, and I think this should affect the overall analysis.

3. Figure 2c shows a higher cluster density for GM12878, while the other two don't differ too much. Is this because of the sequencing depth difference among the Hi-C experiments? Are the Hi-C data downsampled or normalized before 3D model building to reduce experimental biases? How does the DSB/SSB yield/pattern vary as a function of spatial positioning in the nucleus relative to the periphery?

4. The pseudo-random geometry provides a good control to shows the overall importance of 3D geometries generated from Hi-C data (Figure 3b), but this is missing from figure 4b-c where they show induction of DSBs, SSBs/Gy as a function of energy and also in Fig S4. A baseline is always required to understand the improvement. Figure 3 isn't referred within the main text. What is "backbones per DSB" and its importance described in Figure S1? Why isn't it dropping sharply as DSB/Gy? It isn't easy for a reader to understand a plot if the same axis definition varies across the paper. The cluster density in the manuscript can be defined as per number the beads or DSBs within a cluster radius. I would suggest the authors define the Y-axis clearly as either "bead cluster density" or "DSB cluster density" at appropriate places. What is "DSB interchromosome cluster density", where is it defined? Looking just at the numbers from Fig S4, S5, and Fig 5, it is clear that intrachromosomal DSB cluster density plays a major role and has more weightage then interchromosomal DSB cluster density. But the paper lacks the separate analysis of the intrachromosomal DSB part and its difference among cell types. Figure S4 "Carbon 1034.69Kev" shows a completely flattened IMR90-ellipsoid model; why is that?

Reviewer #2: In this manuscript, Ingram, Henthorn et al. present a computational method (G-NOME) for solving 3D structure of the DNA in cell nucleus, based on Hi-C data at the level of topologically associated domains (TADs). The method allows for defining additional constraints to promote lamina-associated domains (LADs) towards the periphery of the solved geometry, and to solve the nucleus shape as an ellipsoid. Using this method and publicly available Hi-C and LAD data, the authors show that derived geometries for different cell types have varying amounts of interchromosomal intermingling. They further simulate radiation-induced double-strand DNA breaks (DSBs) and show that the resulting DSB interchromosome clustering significantly differs between cell types.

The source code of G-NOME is well structured, and released under an open license (GNU GPL), allowing for further developments and reuse. I also found the results very interesting, and I have some suggestions on their presentation, as well as on some methodological aspects. My specific comments are listed below.

Major comments:

1. The target audience of the manuscript includes two broad and separate groups of readers. One of them is knowledgeable in radiation-induced DNA damage, but has only basic knowledge in Hi-C data analysis. The other is familiar with Hi-C data, but has only rudimentary knowledge of radiation-induced DNA damage. The Introduction could be expanded a bit, so that the manuscript would be more accessible to the latter group. In particular, the concepts of DSBs and interchromosonal translocations should be introduced there.

2. A central term in the paper is "cluster density" (CD). It is locally defined as the number of objects that fall within a given radius, and then averaged for all objects in the simulation. This term is closely related (precisely speaking, proportional) to Ripley's K function, commonly used in testing spatial point patterns. (K function was introduced in Ripley, B. D. (1979). Tests of “Randomness” for Spatial Point Patterns. Journal of the Royal Statistical Society - Series B (Statistical Methodology), 41(3), 368–374; you can see https://wiki.landscapetoolbox.org/doku.php/spatial_analysis_methods:ripley_s_k_and_pair_correlation_function for a gentle introduction).

I found the term "cluster density" misleading, because it is not a density (it is not normalized by the volume). As far as I am aware, this term is newly introduced in this manuscript. The authors should consider one of the following:

(A) Define "cluster density" as a density (the number of objects that fall within a given radius, divided by the volume of the ball of this radius).

(B) Use Ripley's K function instead.

(C) Use a different word, say "count", instead of "density".

It would be interesting to see what Fig. 1c and 2c look like when plotting the density as defined in (A).

3. Fig. 7 shows the p-values calculated in >600 statistical tests, without accounting for multiple hypothesis testing. This approach is not reliable, because it does not control for Type I error rate. I suggest to use false discovery rate (FDR) method for this purpose. More importantly, I would plot the effect size (e.g. the interchromosomal bead clustering values or ratio thereof) with a color or symbol depending on the (FDR-corrected) statistical significance. The same applies for Fig. S8. Also it is unclear what does the last sentence of Fig. 7 caption ("The value of...") refer to.

4. Radiation-induced DNA damage at the loci of interactions between two homologous chromosomes can also give rise to mis-repair and abnormal karyotype. The manuscript only considered interchromosomal clustering, but the description of raw data allows for including homolog interactions in the cluster density counting. This possibility (and the resulting observations) should be discussed in the manuscript.

Minor comments:

1. Acronyms such as DSB, SSB and LET should be spelled out at first use.

2. Fig. 1 and Fig. 2 show the same type of analysis, but comparing different underlying data (cell types and cell variants). It would be easier to comprehend the Results section if all the technical aspects were discussed along with explaining Fig. 1, so that the text discussing Fig. 2 could focus on the actual results.

3. It is challenging to compare Fig. 1c and Fig. 2c to Fig. 3bc despite them presenting the same thing; log scale should be used consistently.

4. It is nice to have an interactive 3D version of Fig. 5, but in the paper it would be helpful to show some 2D cuts (for a fixed value of LET and for a fixed value of cluster radius). This would allow the reader to get the main message without opening the 3D plots.

5. Fig. 6b is very difficult to read, due to multiple shapes of symbols being hard to distinguish when they cluster together. I propose to use facets here, and plot different cell types/variants separately.

6. It is implied that the beads in the polymer model have different sizes (as seen in Fig. 1 and 2a). The authors should add one or two sentences explaining how the model accounted for that (are the distances between the beads constant?).

7. The MCMC model tries several bead movement types at random. It would be clearer to enumerate these movements types in the Methods.

8. Lines 144-159: Fig. 3 should be referred to, instead of Fig. 2.

**Have all data underlying the figures and results presented in the manuscript been provided?**

Reviewer #1: Yes

Reviewer #2: Yes

PLOS authors have the option to publish the peer review history of their article (what does this mean?). If published, this will include your full peer review and any attached files.

Reviewer #1: No

Reviewer #2: No
---

## [Decision Letter · Decision Letter 1]

5 Oct 2020

Dear Mr Ingram,

Thank you very much for submitting your manuscript "Hi-C implementation of genome structure for in silico models of radiation-induced DNA damage" for consideration at PLOS Computational Biology.

As with all papers reviewed by the journal, your manuscript was reviewed by members of the editorial board and by several independent reviewers. In light of the reviews (below this email), we would like to invite the resubmission of a revised version that directly addresses the remaining concerns of Reviewer 1. 

We cannot make any decision about publication until we have seen the revised manuscript and your response to the reviewer's comments. Your revised manuscript may be sent to reviewer 1 for further evaluation.

Sincerely,

Ferhat Ay, Ph.D

Associate Editor

PLOS Computational Biology

William Noble

Deputy Editor

PLOS Computational Biology

Reviewer's Responses to Questions

**Comments to the Authors:**

Reviewer #1: The authors have improved the manuscript. However, I still have two concerns

1. Figure S5 and S6 show the spatial distribution of DSB yields in different cell types. How the authors calculated the Normalized DSB frequency? Looking at the Normalized DSB figure, it appears that there is no effect of spatial positioning on DSB yield, but the density result speaks otherwise. Also, how the normalized frequency can have a value above 1 (10.14MeV_1034.69KeV_Carbon)?

2. Figure S13 and S14 shows the model quality generated by G-NOME and Chrom3D programs. Both figures show a significant difference in 3D models with the same set-up. Figure S13 shows a complete reversal of the total amount of DNA content placed in the Periphery and Center. Figure S14 shows a lower proximity score for Chrom3D models. It raises a serious concern about the quality of the models.

Reviewer #2: The authors have fully addressed all the points raised by me and the other reviewer. The revised manuscript is convincing and clearly written.

My only remaining comment is a minor detail: line colors in Fig. 3e should be the same as in Fig. 3b-d.

**Have all data underlying the figures and results presented in the manuscript been provided?**

Reviewer #1: Yes

Reviewer #2: Yes

PLOS authors have the option to publish the peer review history of their article (what does this mean?). If published, this will include your full peer review and any attached files.

Reviewer #1: No

Reviewer #2: No
---

## [Decision Letter · Decision Letter 2]

28 Oct 2020

Dear Mr Ingram,

We are pleased to inform you that your manuscript 'Hi-C implementation of genome structure for in silico models of radiation-induced DNA damage' has been provisionally accepted for publication in PLOS Computational Biology.

Best regards,

Ferhat Ay, Ph.D

Associate Editor

PLOS Computational Biology

William Noble

Deputy Editor

PLOS Computational Biology

Reviewer's Responses to Questions

**Comments to the Authors:**

Reviewer #1: All raised issues were adequately addressed.

**Have all data underlying the figures and results presented in the manuscript been provided?**

Reviewer #1: Yes

PLOS authors have the option to publish the peer review history of their article (what does this mean?). If published, this will include your full peer review and any attached files.

Reviewer #1: No

---

## [Editor Report · Acceptance letter]

7 Dec 2020

PCOMPBIOL-D-20-01179R2 

Hi-C implementation of genome structure for *in silico* models of radiation-induced DNA damage

Dear Dr Ingram,

I am pleased to inform you that your manuscript has been formally accepted for publication in PLOS Computational Biology. Your manuscript is now with our production department and you will be notified of the publication date in due course.

With kind regards,

Nicola Davies
